# Administration of oral dosage forms of medicines to children in a resource limited setting

**Abarna Nadeshkumar** [1,2] *, **Gitanjali Sathiadas** [3], **Shalini Sri Ranganathan** [2]

**1** Department of Pharmacy and Pharmaceutical Sciences, Faculty of Allied Health Sciences, University of Sri Jayewardenepura, Nugegoda, Sri Lanka, **2** Department of Pharmacology, Faculty of Medicine, University of Colombo, Colombo, Sri Lanka, **3** Department of Paediatrics, Faculty of Medicine, University of Jaffna, Jaffna, Sri Lanka

☉ These authors contributed equally to this work.
* abarna@sjp.ac.lk

**Data Availability Statement:** The data relevant to this study are available from OSF at https://osf.io/gp9js/?view_only=48a38d05a6cb4f5cb7e39e4dd302e3d5.

## Abstract

### Background

There are many paediatric specific challenges such as lack of age-appropriate dosage forms, inability of young children to swallow tablets and capsules and poor acceptability, during administration of oral dosage forms of medications to children. Parents adopt various methods which they consider best to circumvent this problem. The objective of this study was to describe the administration practice by parents when giving oral dosage forms of medications to children.

### Methods

A descriptive cross-sectional study was conducted to assess the administration practice of 1800 oral dosage forms of medications administered to children under the age of 12 years using validated indicators. A pre-tested interviewer-administered questionnaire given to parents or caregivers was used to collect the necessary data. The data were analysed using descriptive statistics.

### Results

Data from 1800 oral dosage forms was obtained from 663 children. Of the 1287 solid dosage forms, almost one-third were manipulated by parents at the time of giving the medications to children. They were crushed and dissolved in water given to children. In about 17% of instances safety of water was questionable. In 92% of instances, measuring device was found to be inappropriate.

### Conclusion

Administration of oral dosage forms of medications to children is far from ideal and hinders successful use of medications in children.

**Funding:** Author who receieved the award: NA. Grant No: UGC/DRIC/PG/2015(ii)/SJP/01 Funder: University Grants Commision,Sri Lanka. URL: https://www.ugc.ac.lk/ Funder didn't play any role in study design,data collection and analysis , decision to publish or preparation of the manuscript.

**Competing interests:** The authors have declared that no competing interests exist.

## Introduction

Children do not behave like adults when handling medications. Successful pharmacotherapy in children demands age appropriate prescribing, dispensing and administration of medications. Healthcare professionals face challenges when prescribing and dispensing medications to children [1, 2]. Pharmacotherapy in children will not achieve the intended health outcomes if medications are administered incorrectly. Administration of medications to children by oral route is challenged by children's inability to swallow solid dosage forms such as tablets and capsules. To our best of knowledge studies reporting practices adopted by parents to administer medications to their children are scarce. This study was aimed to describe the administration practice of oral dosage forms to children under 12 years.

## Methods

This was part of a large scale study which investigated the rational use of oral dosage forms (ODFs) in children. The World Health Organisation's (WHO) methodology on "how to investigate drug use in health facilities?" was adopted in the survey of administering practice of ODFs of medications in children [3].

A cross-sectional descriptive study was conducted in a teaching hospital. Required data were collected from the paediatric clinics. The study sample was the ODFs of medications administered to children under the age of 12 years. Information on ODFs of medications was obtained from prescriptions dispensed with at least one oral dosage form of medications within a month before the day of data collection.

Data were collected over a period of two years from January 2017 after administrative approval. (January 2017 to January 2019).

### Sample size and sampling technique

The WHO recommends that there should be at least 600 encounters in a cross-sectional survey with a greater number if possible for drug use studies [3]. Since we were studying the ODFs given to children we equalized one ODF of medication as one encounter. We decided to collect data from a sample of 1800 ODFs using the following inclusion and exclusion criteria.

### Inclusion criteria

i. Prescriptions which had been dispensed with at least one oral dosage form of medication within a month before the day of data collection to children under the age of 12 years

ii. Parents/ guardians providing the data were the ones who gave medications to the child at home

iii. Parents/ guardians must have the prescriptions with them at the time of data collection

### Exclusion criteria

i. Prescriptions without oral dosage forms.

ii. Prescriptions dispensed more than 30 days prior to data collection

Systematic sampling was done to select every other child. Starting point (1st/2nd child) was selected at random. The selected child's prescription was screened to see whether it fulfils the inclusion criteria. In the absence, prescriptions of consequent children were screened until a

prescription satisfying the inclusion criteria was identified. Recruitment of sample selection continued until a total of 1800 ODFs were selected. Data were collected from parents/guardians who visited the clinics using an interviewer-administered questionnaire [S1 Table]. Principal investigator interviewed the parent /guardian while they were waiting to meet the prescriber. Additional information on the reason for repeating of ODFs was also collected. Informed written consent from the parents and assent from child was obtained before data collection.

### Indicators

In a prior study, prescribing, dispensing and administration indicators to measure rational use of ODFs to medications had been already developed using modified RAND Corporation / University of California Los Angeles's (RAND/UCLA) method and validated [4]. The administration indicators focus on the issues related to administration of ODFs. The developed administration indicators were converted to an interviewer-administered questionnaire. This was pre-tested and inter-rater reliability was determined. Numerators and denominators to calculate the indicators were also defined. The administration indicators are given below.

1. Percentage of instances where the child swallowed the intact tablet/capsule

2. Percentage of instances where crushed tablet was dissolved and administered

3. Percentage of liquid oral dosage forms administered using an oral syringe

4. Percentage of instances where safe water was used in preparing the medicine

5. Percentage of instances where the prescribed dose is correctly completed

### Data entry and analysis

Data were entered in a custom made database. Accuracy of data entry was independently cross-checked. Descriptive statistics was used to calculate the indicators. Children were categorized based on age. Data from term new born [child born on the day of data collection (0 day)] up to 12 years were collected. They were categorized as term new-born (0–27 days), infants and toddlers (1 month to 23 months) pre-school children (2–5 years) and school children (6–11 years) [5]. Tablets, capsules and powders were considered as solid dosage forms, syrups and suspension were taken as liquid dosage forms. Boiled cooled water and bottled drinking water was considered as safe water. Well water was defined as water from hole drilled into the ground to access water contained in an aquifer [6] and water supplied through a water pump. Tube well water was defined as water obtained through a narrow screened tube, pipe or casing that and driven into a subsurface aquifer [7].

### Approvals

Ethical approval was obtained from ethics review committee, Faculty of Medicine, University of Colombo, Sri Lanka (EC-15-022). Administrative approval was obtained from the Director/ Teaching Hospital and other relevant administrative authorities.

### Results

Data on 1800 ODFs were collected from prescriptions issued to 663 children: Infants and children accounted for 20.5% (N = 136), pre-school children 35.4% (N = 235) and school children 44% (N = 292). Male: female ratio was 1.25.

**Table 1. Type of oral dosage forms administered to children under the age of 12 years.**

| Age category* | Tablet N (%) | Capsule N (%) | Syrup N (%) | Suspension N (%) | Powder** N (%) | Total N (%) |
|---|---|---|---|---|---|---|
| Infants and toddlers | 187 (47.9) | 0 (0) | 183 (46.9) | 14 (3.6) | 6 (1.5) | 390 (100) |
| Preschool children | 395 (60.8) | 1 (0.2) | 228 (35.1) | 23 (3.5) | 3 (0.5) | 650 (100) |
| School age children | 688 (90.5) | 4 (0.5) | 55 (7.2) | 10 (1.3) | 3 (0.5) | 760 (100) |
| Total | **1270 (70.5)** | **5 (0.3)** | **466 (25.9)** | **47(2.6)** | **12 (0.7)** | **1800 (100)** |

*Children were categorized based on age as term new-born (0–27 days), infants and toddlers (1 month to 23 months) pre-school children (2–5 years) and school children (6–11 years)

**Powders were extemporaneously prepared and dispensed by the pharmacist at the pharmacy

Percentages may not sum to 100% due to rounding of numbers

Tablets (70.6%) followed by syrups (25.9%) were the most commonly prescribed ODFs for this cohort of children (Table 1). Tablets accounted for 47.9% ODF prescribed for infants and toddlers compared to 76.8% in older children. This was reversed for syrups: 46.9% for infants and toddlers compared to 20% for older children.

## Indicator 1: Percentage of instances where the child swallowed the intact tablet/ capsule

The above indicator was calculated to be 57% (95% CI: 54% -60%). Required data were obtained from 1287 solid ODFs.

Out of 552 oral dosage forms that were not swallowed intact 442 [80% (95% CI: 76%-83%] were given to children under the age of six. As shown in Fig 1, 250 of the 552 ODFs that were not taken as intact tablet and were taken solely due to the administration difficulty to swallow the whole tablet. They were either split or crushed. Out of the manipulated tablet/capsule 22% were with narrow therapeutic index.

## Indicator 2: Percentage of instances where crushed tablet was dissolved and administered

The above indicator was calculated to be 32% (95% CI: 29%- 34%). Required data to calculate this indicator were obtained from 1287 solid ODFs. Out of 411 ODFs which were crushed 12 ODFs were crushed by pharmacist when dispensing and others by the parent /caregiver at the home setting. Hydrocortisone (50%) and propanol (25%) were two oral dosage forms that

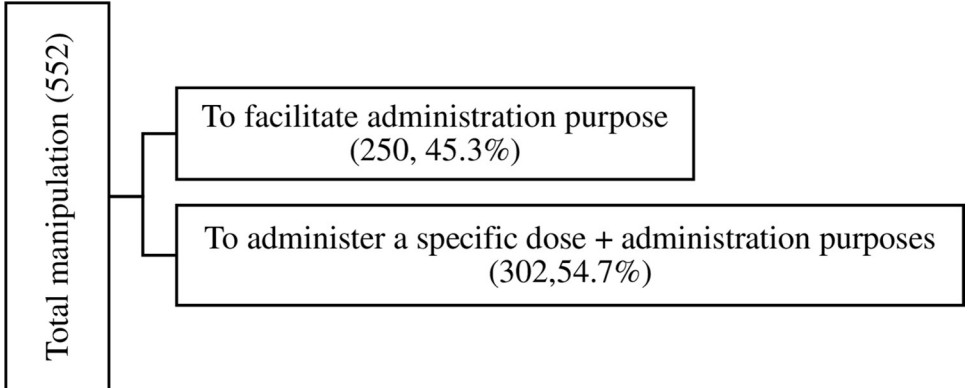

**Fig 1. Reasons for administering manipulated ODFs children.**

were frequently crushed and dispensed as powder sachets by pharmacists whereas vitamins (51%, 95% CI: 47%-56%) were frequently crushed by the parents /caregivers. Parents/ caregivers also had to crush tablets of paracetamol, carbamazepine, sodium valproate, hydrochlorothiazide, thyroxin, hydrocortisone and fludrocortisone.

Crushed tablets were dissolved in water (80.3%), milk (14.4%), other liquid medications (4.1%) and other liquids (1.2%).

### Indicator 3: Percentage of liquid oral dosage forms administered using an oral syringe

The above indicator was calculated to be 8% (95% CI: 6%-11%). Required data to calculate this indicator were obtained from 513 liquid ODFs. Commercially available measuring cups followed by the droppers were the commonly used measuring devices to give liquid dosage forms (Fig 2). Only less than 1% has used household spoons.

### Indicator 4: Percentage of instances where safe water was used in preparing the medicine

The above indicator was calculated to be 83% (95% CI: 78%-87%). Data to calculate this indicator were obtained from 330 instances where water was required to prepare the medications. The rest of the instances tube well water (2.4%) and well water (14.5%) was used.

### Indicator 5: Percentage of instances where the prescribed dose is correctly completed

The above indicator was calculated to be 98.8% [95% (CI; 98%-99%)]. Required data to calculate this indicator were obtained from 1800 ODFs.

The summary of the indicator values is given in Table 2.

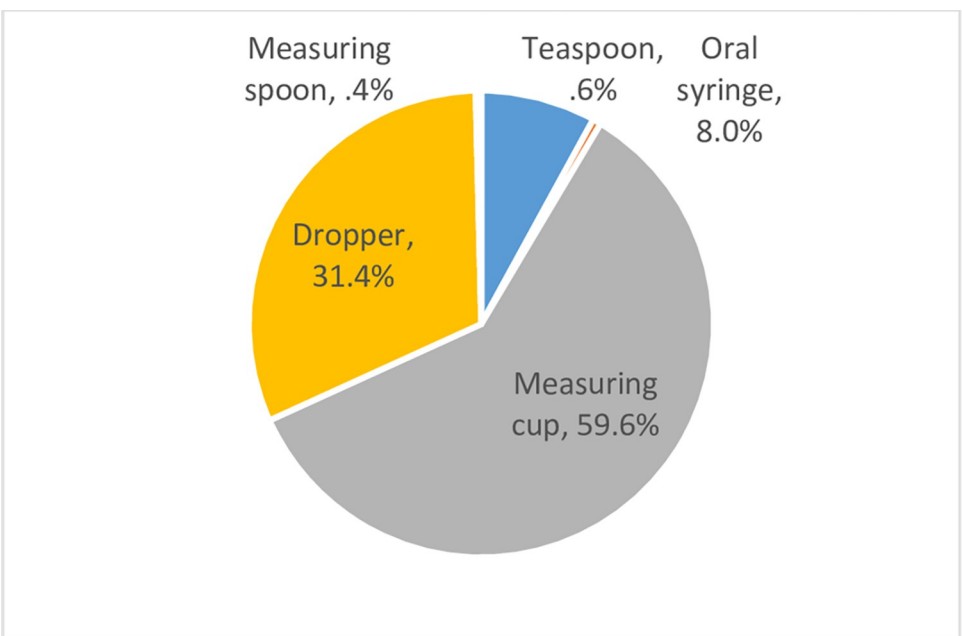

**Fig 2. Measuring devices used by children under the age of 12 years to administer liquid dosage forms.**

**Table 2. Summary of the indicator values.**

| | Indicator | Ideal value (%) | Calculated value (%) |
|---|---|---|---|
| 1 | Percentage of instances where the child swallowed the intact tablet/capsule | 100 | 57 |
| 2 | Percentage of instances where crushed tablet was dissolved and administered | 0 | 32 |
| 3 | Percentage of liquid oral dosage forms administered using an oral syringe | 100 | 8 |
| 4 | Percentage of instances where safe water was used in preparing the medicine | 100 | 83 |
| 5 | Percentage of instances where the prescribed dose is correctly completed | 100 | 98.8 |

*Reason for repeating the dose.* Out of 115 instances where the dose was repeated the child spitting in the first instances was 83%. This was observed five times more in children under the age of 6 years compared to school children.

## Discussion

This study has explored the challenges encountered during administration of ODFs medications to young children in a resource-limited country. Furthermore, we were able to document the prevalence of the rational use of ODFs in a large scale study which is expected to add to the limited evidence available in the literature. The majority of the studies in the literature have described the challenges with no uniform indicators [8, 9]. However, in this study, we were able to document the challenges in an objective way facilitating repeat studies after any form of intervention and comparing the status with the other settings.

A little under half (43%) of the children in our study did not swallow the solid ODFs intact. This was almost in agreement with Tanzanian study (52%) [10]. Due to medications available as unsuitable dosage forms, parents and pharmacists are compelled to manipulate the solid ODFs frequently. Even if the medications are available in suitable dosage forms, they had to be split or crushed because most of them are available only in adult size. The problems documented in our study such as crushing/breaking pill, swallowing the dry powder from crushed pill, drinking crushed/dissolved pill mixed with water appears to be occurring in other resource limited countries as well [11, 12]. Splitting or crushing the tablets poses the risk of administering an inaccurate dose, which could contribute to either over or under dosing. However, it has the potential to cause significant clinical outcome if the drug concern has a narrow therapeutic index. Several studies have shown that quality of such manipulated product is unpredictable [13, 14] There are instances where young children were given manipulated solid dosage forms despite the availability of liquid dosage forms. Reasons for such practice were cost minimisation and children refusing to take liquid dosage forms because of poor palatability. This was documented by other researchers also [15]. Parents, therefore, resort to manipulate the solid dosage forms rather than giving liquid medicines to children, without realizing the problems related to stability, bioavailability and dose accuracy when solid dosage forms are manipulated [16]. This is one of the reason why flexible solid dosage forms are now been promoted as suitable dosage form for young children.

Though oral syringes are recommended when oral liquid dosage forms are prescribed in doses other than multiples of 5 mL [17] this is not regularly practised in our country, compelling the parents to use measuring cups or spoons. It was observed that the pharmacists were also providing measuring cups when dispensing liquid medicines. Even though measuring cups and spoons are better than domestic measuring devices, studies have reported that even

pharmacists were not able to measure the dose accurately with measuring cups or spoons. Regulatory authorities should demand that an oral syringe to be provided with liquid dosage forms of medicines which could be prescribed in doses other than multiples of 5 mL. This is critical for medications with narrow therapeutic index where small inaccuracy in doses could lead to toxicity or therapeutic failure and antibacterial agents which should be in a consistent steady state concentration at the site of infection.

Drinking unclean water puts children at risk. As children's bodies are not as strong as adults, drinking unsafe water puts children in danger of deadly bouts of diarrhoea, parasites and other water related diseases [18]. The use of safe water in our survey was found to be better (75%) than the Tanzanian (41%) survey [10]. This indicator can be further improved by educating the parents. In agreement with several studies in the literature the problems of ODF tends to be high in young children (under 6 years) [19, 20]. The reasons are numerous: children's growing awareness, children's swallowing ability. Since these children may not be able to understand the medications and disease we recommend parental education is important. This can take in the form of posters which can be placed in child clinics so that parents can read and understand during the waiting time. Parents could be given illustrated information leaflets with instructions and pictures to keep with them and practice when the need arises.

The study was limited to children who were receiving treatment from a single large hospital. Hence, results may not be generalizable for the entire country. However, it is highly unlikely that the indicators would be higher than what we have reported from a teaching hospital. Repeating this study in other settings would confirm our findings while also documenting the challenges and exploring the areas that were not covered in this study. We have limited the study to calculate the indicators in order to highlight the incorrect administration practices in paediatric pharmacotherapy, hence we did not look at demographics, adherence, number of children in the household and potential determinants of correct administration practice. Further studies should be done to explore the predisposing factors for inappropriate administration of medications to children by parents.

## Conclusion

Administration practices of ODFs of medications to children have room for improvement. Parents are compelled to manipulate the available dosage forms due to required dose and administration difficulty. Necessity of age appropriate dosage forms needs to be re-looked by the authorities. Even though oral syringes are used in developing countries to administer liquid dosage forms the use is very less in this setting. Oral syringes should be provided at least to children who are on medications with narrow therapeutic index. Overdosing the medications with narrow therapeutic index can result in life-threatening adverse drug events, whereas under dose errors lead to decreased therapeutic effects. Safe water is accessible to most of the children. Healthcare professionals need to be aware of the risk of wrong administration practices by parents /caregivers of children and educate them.

## Supporting information

**S1 Table. Data collection sheet-administration indicators.**
(DOCX)

## Author Contributions

**Conceptualization:** Abarna Nadeshkumar, Gitanjali Sathiadas, Shalini Sri Ranganathan.

**Data curation:** Abarna Nadeshkumar.

**Formal analysis:** Abarna Nadeshkumar.

**Funding acquisition:** Abarna Nadeshkumar.

**Investigation:** Abarna Nadeshkumar.

**Methodology:** Shalini Sri Ranganathan.

**Project administration:** Abarna Nadeshkumar.

**Resources:** Abarna Nadeshkumar.

**Supervision:** Gitanjali Sathiadas, Shalini Sri Ranganathan.

**Writing – original draft:** Abarna Nadeshkumar.

**Writing – review & editing:** Gitanjali Sathiadas, Shalini Sri Ranganathan.

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
