## [Decision Letter · Decision Letter 0]

6 Apr 2022

PONE-D-21-27919Administering of oral dosage forms of medicines to children in a resource limited settingPLOS ONE

Dear Dr. Nadeshkumar,

Thank you for submitting your manuscript to PLOS ONE. After careful consideration, we feel that it has merit but does not fully meet PLOS ONE’s publication criteria as it currently stands. Therefore, we invite you to submit a revised version of the manuscript that addresses the points raised during the review process.

We look forward to receiving your revised manuscript.

Kind regards,

Vijayaprakash Suppiah, PhD

Academic Editor

PLOS ONE

https://journals.plos.org/plosone/s/file?id=ba62/PLOSOne_formatting_sample_title_authors_affiliations.pdf".

Reviewers' comments:

Reviewer's Responses to Questions

**Comments to the Author**

1. Is the manuscript technically sound, and do the data support the conclusions?

Reviewer #1: Partly

2. Has the statistical analysis been performed appropriately and rigorously? 

Reviewer #1: No

3. Have the authors made all data underlying the findings in their manuscript fully available?

Reviewer #1: No

4. Is the manuscript presented in an intelligible fashion and written in standard English?

Reviewer #1: No

5. Review Comments to the Author

Reviewer #1: Thank you for the opportunity to review your manuscript regarding safe and appropriate administration of oral formulations to infants and young children.

Here are my comments:

1. English is a tricky language. Your manuscript would be greatly improved by having a native English speaker review and revise the wording and sentence structure. For example, administration vs administering is the appropriate word in the title;

Consider changing to: Administration of oral medication dosage forms to children in a limited resource setting (better flow in common English language)

2. Please define all abbreviations upon first use e.g. ODF, and terms such as safe water, well water, and tube well water in the first part of the paper and risks of unsafe water to dissolve oral medications.

4. The procedure for selection of the prescriptions for review in the study is unclear. This section requires revision with clear inclusion and exclusion criteria, listing the dates of the study period, and ages of patients 0- 12 years?

Was the medication prescribed for a minimum of 1 month or < 1 month? Please clarify in procedure.

5. Was adherence to the prescription evaluated? - goal of oral dosage forms that can be easily administered and tolerated, correct?

6. Was there a difference in patient practice if they had one child vs > 1 child? Would be interesting to compare these groups

7. Were all oral medications included in this study able to be safely crushed? Please refer to ISMP list Medications that should not be crushed: updated in 2021 https://www.ismp.org/recommendations/do-not-crush

8. Did you collect demographics beyond age of children? How about diagnoses, area of residence (urban vs rural?), etc

9. Moving beyond descriptive statistics, consider regressions examining factors associated with appropriate oral dosage administration for a more meaningful impact on practice. Which authors were involved in the analysis?

10. Discussion, regarding need for printed education for parents (is there a literacy issue?), how about "teach back" method

6. PLOS authors have the option to publish the peer review history of their article (what does this mean?). If published, this will include your full peer review and any attached files.

Reviewer #1: **Yes: **Victoria Tutag Lehr, Pharm D Professor Pharmacy Practice Eugene Applebaum College of Pharmacy & Health Sciences, Wayne State University, Detroit MI 48201

---

## [Author Response · Author response to Decision Letter 0]

15 Jun 2022

Dear Editor, 

We would like to thank you and the reviewer for careful and thorough reading of this manuscript and for the comments and positive suggestions. Our response to each point raised are given below and the corrections are highlighted in the revised manuscript.

Thanking you. 

Yours Sincerely,

Abarna Nadeshkumar

(Corresponding author) 

Response from authors to the comments 

1) Responses to the comments from Academic Editor 

Author’s Response: Thank you very much. We apologize for not following the journal style. We have now revised the whole manuscript based on the journal style

Author’s Response: The copy of the questionnaire is given as supporting information

Author’s Response: Thank you and the data set will be made available as requested if accepted for publication.

Author’s Response: Thank you and the data set will be made available as requested if accepted for publication.

2) Responses to the comments from Reviewer 

1. Reviewer Comment: English is a tricky language. Your manuscript would be greatly improved by having a native English speaker review and revise the wording and sentence structure. For example, administration vs administering is the appropriate word in the title;

Consider changing to: Administration of oral medication dosage forms to children in a limited resource setting (better flow in common English language)

Author’s Response: Taking into consideration reviewer´s comment, we thoroughly revised the manuscript and corrected the manuscript. 

We have amended the term administering to administration and we have amended the title also as suggested 

2. Reviewer Comment : Please define all abbreviations upon first use e.g. ODF, and terms such as safe water, well water, and tube well water in the first part of the paper and risks of unsafe water to dissolve oral medications.

 Author’s Response: 

I. Abbreviations are defined in the first use (Line numbers 56, 57, 95) and terms are defined in the data analysis section. 

II. Definitions for the terms are given below.(Added in the data analysis section )

Well water- water from hole drilled into the ground to access water contained in an aquifer. Water is supplied through a water pump (Line number 115) 

Tube well water- water obtained through a narrow screened tube, pipe or casing that is driven into a subsurface aquifer. (Line number 116)

III. Risks of unsafe water to dissolve oral medications in given in the discussion. (Line number 226 ) 

3. Reviewer Comment: The procedure for selection of the prescriptions for review in the study is unclear. This section requires revision with clear inclusion and exclusion criteria, listing the dates of the study period, and ages of patients 0- 12 years?

Author’s Response: Thank you. Inclusion exclusion criteria and the data study period have been added to the manuscript.

1) Inclusion exclusion criteria (Line number 73 under methods)

Inclusion criteria

I. Prescriptions which had been prescribed with at least one oral dosage form of medicine within a month from the day of data collection to children under the age of 12 years

II. Parents/ guardians giving the data were the ones who gave medicines to the child at home

III. Parents/ guardians should have the prescriptions with them at the time of data collection.

Exclusion criteria

I. Prescriptions without oral dosage forms.

II. Prescriptions issued. older than one month

2) Dates of study period (Line number 66 under methods)

January 2017 to January 2019 

3) Ages of patients- Line number 110

Data from term new born [child born on the day of data collection (0 day)] up to 12 years were collected. They were categorized as term new-born (0-27 days), infants and toddlers (1 month to 23 months) pre-school children (2-5 years) and school children (6-11 years)

Reviewer Comment: Was the medication prescribed for a minimum of 1 month or < 1 month? Please clarify in procedure.

Author’s Response: Data were collected when the prescription was prescribed within one month before the day of data collection. Eg if the data was collected on 1st February 2017, prescriptions prescribed between 1st January 2017 to 31st January 2017 were considered. (Line number 63)

4. Reviewer Comment: Was adherence to the prescription evaluated? - goal of oral dosage forms that can be easily administered and tolerated, correct?

Author’s Response: We regret to inform you that we did not evaluate the adherence to prescriptions however we evaluated whether the dose was completed or not and the reasons for repeating the dose.

5. Reviewer Comment: Was there a difference in patient practice if they had one child vs > 1 child? Would be interesting to compare these groups

Author’s Response: Sorry, we didn’t not collect the data on number of children 

6. Reviewer Comment: Were all oral medications included in this study able to be safely crushed? Please refer to ISMP list Medications that should not be crushed: updated in 2021 https://www.ismp.org/recommendations/do-not-crush

Author’s Response: ISMP list is given in brand names which are mostly not available in Sri Lanka. Unfortunately we collected only the generic names during the data collection. Brand names of the medicines administered by parents at the time of data collection were not known. In resource limited countries like Sri Lanka due to non-availability of required strength and age appropriate dosage forms for young children, health care professionals and parents are compelled to crush the available adult tablet without knowing any data on bioavailability, safety, palatability and dose accuracy (Somasiri, U., Thillainathan, S., Fernandopulle, R. and Sri Ranganathan, S., 2013. Antiepileptic drugs for children: Availability, suitability and acceptability. Sri Lanka Journal of Child Health, 42(1), pp.38–39 , Hoppu K, Sri Ranganathan S, Dodoo AN. Realities of paediatric pharmacotherapy in the developing world. Arch Dis Child. 2011 Aug;96(8):764-8. doi: 10.1136/adc.2009.180000. Epub 2011 Mar 25. PMID: 21441240.) 

7. Reviewer Comment: Did you collect demographics beyond age of children? How about diagnoses, area of residence (urban vs rural?), etc

Author’s Response: No, we did not collect demographics beyond age

8. Reviewer Comment: Moving beyond descriptive statistics, consider regressions examining factors associated with appropriate oral dosage administration for a more meaningful impact on practice. Which authors were involved in the analysis?

Author’s Response: We agree with your comments. Regression analysis would have greatly improved our paper, however examining the factors associated with appropriate oral dosage administration was not an objective of this paper. This study was aimed to describe the administration practices using the indicators .We did not focus on the factors affecting the practices in this study. After documenting the extend of problem we are in the process developing a further study to look in to the factors.

First author was involved in the analysis.

14. Reviewer Comment: Discussion, regarding need for printed education for parents (is there a literacy issue?), how about "teach back" method

Author’s Response: Luckily, there is no literacy issues with regard to native language in Sri Lanka. The literacy rate of Sri Lanka is above 90%. In our personnel experiences we have noticed that both the doctors and the pharmacists as well as the parents don’t have time to give or listen to advices. The clinics and the pharmacies are usually overcrowded and the waiting time to get the service is long. Hence we hypothesized that printed education materials which can be displayed in the clinics and distributed to parents while they are waiting to receive the service will be a suitable intervention

---

## [Decision Letter · Decision Letter 1]

4 Jul 2022

PONE-D-21-27919R1Administration of oral dosage forms of medicines to children in a resource limited settingPLOS ONE

Dear Dr. Nadeshkumar,

Thank you for submitting your manuscript to PLOS ONE. After careful consideration, we feel that it has merit but does not fully meet PLOS ONE’s publication criteria as it currently stands. Therefore, we invite you to submit a revised version of the manuscript that addresses the points raised during the review process.

We look forward to receiving your revised manuscript.

Kind regards,

Vijayaprakash Suppiah, PhD

Academic Editor

PLOS ONE

Journal Requirements:

Reviewers' comments:

Reviewer's Responses to Questions

**Comments to the Author**

1. If the authors have adequately addressed your comments raised in a previous round of review and you feel that this manuscript is now acceptable for publication, you may indicate that here to bypass the “Comments to the Author” section, enter your conflict of interest statement in the “Confidential to Editor” section, and submit your "Accept" recommendation.

Reviewer #1: (No Response)

2. Is the manuscript technically sound, and do the data support the conclusions?

Reviewer #1: Partly

3. Has the statistical analysis been performed appropriately and rigorously? 

Reviewer #1: No

4. Have the authors made all data underlying the findings in their manuscript fully available?

Reviewer #1: No

5. Is the manuscript presented in an intelligible fashion and written in standard English?

Reviewer #1: No

6. Review Comments to the Author

Reviewer #1: Thank you for allowing me to review this revised manuscript. While much improved there are some persistent areas that require attention and revision:

1)Inclusion exclusion criteria (Line number 73 under methods)

Inclusion criteria

I.Prescriptions which had been prescribed with at least one oral dosage form of

medicine within a month from the day of data collection to children under the age of 12

years (dispensed may be a more definite word here vs prescribed?) Medication is the usual term vs medicine

II.Parents/ guardians giving the data were the ones who gave medicines to the child at

home (giving suggest change to providing)

III.Parents/ guardians should have the prescriptions with them at the time of data

collection. (recommend change "should" to must to strengthen the inclusion criteria)

Exclusion criteria

I.Prescriptions without oral dosage forms.

II.Prescriptions issued. older than one month (recommend change this sentence-doesnt make sense in current form. Suggest something similar to "Prescriptions dispensed more than 30 days prior to data collection"

Please re-review the entire manuscript carefully for correct grammar and typos

Again, suggest review by a native English speaker, perhaps not involved in the study to obtain a clear concise edited text

There are persistent grammatical and a spelling error

Please see lines 72, 194,200,206, 226,228 for revision

With respect to regression analysis, this recommendation was rejected based on that this was not in the original study question, however it may strengthen the study. If you do not wish to run the regression to provide a robust analysis of this issue, then please discuss the need to explore factors predisposing to inappropriate administration of medications by parents to children in your discussion. Thank you.

Clearly state in the Discussion section exactly how information to parents/caregivers in a busy clinic with long wait times can be provided (printed information on posters, handouts, etc)

Recommend stating clearly in a section on limitations that demographics, adherence and number of children in the household were not evaluated

When writing about the incorret use of medication dispensing devices and drugs of narrow therapeutic index, please specify the proportion of your study drugs where of this nature...for example in a regional referral center, children may receive anticoagulants, antiarryhmics, antiepileptics that require exact dosage administration

Finally, in the conclusion when stating issues, strongly recommend that some of these are stated...such as disease transmission, bioavailability, and over and underdosing. The current conclusion is rather weak.

Good luck in your final revisions!

7. PLOS authors have the option to publish the peer review history of their article (what does this mean?). If published, this will include your full peer review and any attached files.

Reviewer #1: No

---

## [Author Response · Author response to Decision Letter 1]

6 Sep 2022

1. Reviewer’s Comment: Thank you for allowing me to review this revised manuscript. While much improved there are some persistent areas that require attention and revision:

1)Inclusion exclusion criteria (Line number 73 under methods)

Inclusion criteria

I. Prescriptions which had been prescribed with at least one oral dosage form of

medicine within a month from the day of data collection to children under the age of 12

years (dispensed may be a more definite word here vs prescribed?) Medication is the usual term vs medicine

Author’s response: Thank you for the comment. I have changed the word prescription into dispensing (Line number 74) and the word medicine in to medication (Entire manuscript) 

2. Reviewer’s Comment: Parents/ guardians giving the data were the ones who gave medicines to the child at home (giving suggest change to providing)

Author’s response: Thank you for the comment. The wording has been changed to providing (Line number 76).

3. Reviewer’s Comment: Parents/ guardians should have the prescriptions with them at the time of data collection. (recommend change "should" to must to strengthen the inclusion criteria)

Author’s response: The wordings has been changed to must (Line number 78) .

4. Reviewer’s Comment: Exclusion criteria

I. Prescriptions without oral dosage forms.

II. Prescriptions issued. Older than one month (recommend change this sentence-doesn’t make sense in current form. Suggest something similar to "Prescriptions dispensed more than 30 days prior to data collection"

Author’s response: The recommended change has been done (Line number 82).

5. Reviewer’s Comment: Please re-review the entire manuscript carefully for correct grammar and typos

Again, suggest review by a native English speaker, perhaps not involved in the study to obtain a clear concise edited text

There are persistent grammatical and a spelling error

Please see lines 72, 194,200,206, 226,228 for revision

Author’s response: Entire manuscript was re reviewed for grammar and typos and the lines 74,200,205,211,233,235 were revised.

6. Reviewer’s Comment: With respect to regression analysis, this recommendation was rejected based on that this was not in the original study question, however it may strengthen the study. If you do not wish to run the regression to provide a robust analysis of this issue, then please discuss the need to explore factors predisposing to inappropriate administration of medications by parents to children in your discussion. Thank you.

Author’s response: Thank you for your comment. It has been discussed in the discussion (Line number 250).

7. Reviewer’s Comment: Clearly state in the Discussion section exactly how information to parents/caregivers in a busy clinic with long wait times can be provided (printed information on posters, handouts, etc)

Author’s response: Thank you for your comment. It has been discussed in the discussion (Line number 237).

8. Reviewer’s Comment : Recommend stating clearly in a section on limitations that demographics, adherence and number of children in the household were not evaluated

Author’s response: Limitation has been clearly stated in the discussion now. (Line number 246).

9. Reviewer’s Comment: When writing about the incorrect use of medication dispensing devices and drugs of narrow therapeutic index, please specify the proportion of your study drugs where of this nature...for example in a regional referral center, children may receive anticoagulants, antiarryhmics, antiepileptics that require exact dosage administration

Finally, in the conclusion when stating issues, strongly recommend that some of these are stated...such as disease transmission, bioavailability, and over and under dosing. The current conclusion is rather weak.

Good luck in your final revisions!

Author’s response: Proportion of the study drug has been mentioned (Line numbers 145,155 .The conclusion has been rewritten. (Line number 252

---

## [Decision Letter · Decision Letter 2]

6 Oct 2022

Administration of oral dosage forms of medicines to children in a resource limited setting

PONE-D-21-27919R2

Dear Dr. Nadeshkumar,

We’re pleased to inform you that your manuscript has been judged scientifically suitable for publication and will be formally accepted for publication once it meets all outstanding technical requirements.

Kind regards,

Vijayaprakash Suppiah, PhD

Academic Editor

PLOS ONE

Reviewers' comments:

Reviewer's Responses to Questions

**Comments to the Author**

1. If the authors have adequately addressed your comments raised in a previous round of review and you feel that this manuscript is now acceptable for publication, you may indicate that here to bypass the “Comments to the Author” section, enter your conflict of interest statement in the “Confidential to Editor” section, and submit your "Accept" recommendation.

Reviewer #1: (No Response)

2. Is the manuscript technically sound, and do the data support the conclusions?

Reviewer #1: Yes

3. Has the statistical analysis been performed appropriately and rigorously? 

Reviewer #1: Yes

4. Have the authors made all data underlying the findings in their manuscript fully available?

Reviewer #1: No

5. Is the manuscript presented in an intelligible fashion and written in standard English?

Reviewer #1: No

6. Review Comments to the Author

Reviewer #1: Thank you again for the opportunity to review your second version of edits

Overall, the much improved and I recommend that the manuscript be accepted for publication with two minor grammatical revisions:

Line 256: re-looked -a better word may be "re-examined"

Line 257: very less - please change to "less"

(do not require the "very")

Good luck with your future work

7. PLOS authors have the option to publish the peer review history of their article (what does this mean?). If published, this will include your full peer review and any attached files.

Reviewer #1: No

---

## [Editor Report · Acceptance letter]

14 Dec 2022

PONE-D-21-27919R2 

Administration of oral dosage forms of medicines to children in a resource limited setting 

Dear Dr. Nadeshkumar:

I'm pleased to inform you that your manuscript has been deemed suitable for publication in PLOS ONE. Congratulations! Your manuscript is now with our production department. 

Kind regards, 

on behalf of

Dr. Vijayaprakash Suppiah 

Academic Editor

PLOS ONE